# Performance Analysis of Soft-Switching FSO/THz-RF Dual-Hop AF-NOMA Link Based on Cognitive Radio

Rongpeng Liu [1,2], Ziyang Wang [1,2], Xuerui Wang [1,2], Jingwei Lu [1,2], Yawei Wang [1,2], Yizhou Zhuo [1,2], Ruihuan Wu [1,2], Zhongchao Wei [1,2] and Hongzhan Liu [1,2,*]

1   Guangdong Provincial Key Laboratory of Nanophotonic Functional Materials and Devices, Guangzhou 510006, China; lrp2021@m.scnu.edu.cn (R.L.)
2   School of Information and Optoelectronic Science and Engineering, South China Normal University, Guangzhou 510006, China
*   Correspondence: hongzhanliu@m.scnu.edu.cn

**Abstract:** This paper presents a promising solution to address the scarcity of spectrum resources and enhance spectrum efficiency in the context of cognitive radio (CR)-based soft-switching free-space optical (FSO)/terahertz (THz) radio frequency (RF) dual-hop amplify-and-forward (AF)–non-orthogonal multiple access (ROMANO) links. The impact of maximum tolerable interference power in the primary network, transmit power in the secondary transmitter, and maximum relay transmission power on the link are thoroughly studied. The numerical results ultimately validate the effectiveness of this link in improving performance, and a comparative analysis is conducted with the without-CR scheme, highlighting the distinctive characteristics of the proposed link.

**Keywords:** free-space optics (FSO); terahertz (THz); radio frequency (RF); non-orthogonal multiple access (NOMA); cognitive radio (CR)



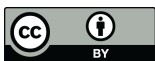

## 1. Introduction

Wireless communication has attained the utmost importance in the digital era, and the advent of the sixth-generation (6G) wireless networks has further underscored its significance [1]. However, scarce spectrum resources [2] and low spectrum efficiency (SE) [3] are two main problems that really need to be solved in radio frequency (RF) communication immediately. Despite this, the RF band is still a better choice for connecting with users because of its better bypassing capabilities [4]. Therefore, a link that addresses scarce spectrum resources and improves SE is worth investigating.

Free-space optics (FSO) and terahertz (THz) have become one of the key technologies to address the scarce spectrum resources [5]. On the one hand, FSO provides a high available bandwidth and high-speed transmission rates, and can effectively mitigate the current shortage of RF communication bands [6]. On the other hand, THz with a frequency of 0.1–10 THz have a high frequency in order to meet the requirements of the ultra-dense small cell networks, which facilitates the high data rate coverage and secure transmission in the wireless network [7]. These unique advantages have made FSO and THz indispensable in overcoming the limitations of the RF bands.

It is worth noting that FSO and THz resources have no limitations compared to RF, but both are affected by atmospheric conditions during propagation. Specifically, FSO outperforms THz in clear weather, while THz is more resistant to fog, dust, and air turbulence (scintillation only) than FSO. These effects are due to the presence of water vapor, scattering of particles of different sizes, and changes in the refractive index of the air [8]. As a result, FSO and THz are suitable as a hybrid link for high-speed transmission. In the research of hybrid link switching, two main types are currently studied: hard-switching and soft-switching. In hard-switching, both FSO and THz have only one threshold; when the transmitted signal-to-noise ratio (SNR) is higher than the threshold of one link, the link can

be activated, if it is higher than the threshold of both at the same time, the corresponding link can be activated according to the preset situation, and if they are both lower than the threshold of both, the hybrid link will be outage. For soft-switching, FSO requires two thresholds to be set, while THz requires only one threshold. Therefore, THz links work the same as when hard-switching, and for two threshold FSO links, the relationship with the high and low thresholds is judged. Similarly, if both are outage, the hybrid link is outage. The proposed soft-switching hybrid link increases the complexity of the link, but it greatly improves the problem of frequent switching damage to the link by hard-switching [9].

The proposal of using non-orthogonal multiple access (NOMA) is a revolutionary solution to the issue of low SE [10]. By enabling multiple users to share frequency bands or time bands, NOMA increases the communication capacity in the new generation of wireless networks [11]. While NOMA can undoubtedly enhance SE, it still requires the allocation of a specific frequency band. This becomes particularly adverse in an age where spectrum usage remains disappointingly low. For instance, in 2005, the spectrum usage in Chicago for the 30–3000 MHz range stood at 17.4% [12]. In 2008, the spectrum usage for the 80–5850 MHz range in Singapore was 4.54% [13]. Similarly, in 2012, the spectrum usage for the 30–3000 MHz range in Beijing was 7.63% [14].

In order to solve the abovementioned problem of low spectrum usage, cognitive radio (CR)–NOMA has been proposed. There are three ways of spectrum sharing based on CR: underlay, interweave, and overlay [15]. In the case of underlay CR, the secondary transmitter (ST) and the primary transmitter (PT) may transmit simultaneously using the original frequency bands of the primary network (PN), ensuring that the interference from the secondary network (SN) to the PN is below a tolerance level. This way needs to protect the communication quality of the PN, and it has strict requirements for spectrum sharing. In interweave CR, the cognitive engine first determines the frequency bands that the PN has a license to use, and when the PN does not detect activity in these bands, the SN uses these licensed bands. The determination of these spectrum holes by the cognitive engine is called spectrum sensing. This way requires the flexibility to detect the spectrum usage of the PN at any time. In the case of overlay CR, the ST is transmitted simultaneously with the PT, but the SN needs to be used as a relay to forward the signals from the primary user (PU) to the target receiver to compensate for the interference caused to the PN. This way requires the SN to bear part of the signals' transmission in the PN. This increases the complexity of the SN. The fusion of underlay CR and NOMA has received particular attention in the last few years due to the fact that underlay CR, compared to interweave CR, can be transmitted simultaneously with the PU, increasing the flexibility of the SU, and not being responsible for the signals of the PU reduces the burden on the SN, compared to overlay CR [16].

By incorporating CR into the SN, underutilized spectrum bands in the PN can be effectively utilized, which not only improves SE and optimizes spectrum usage [17] but also contributes to a more efficient allocation of limited spectrum resources [18]. Leveraging CR, the SN can access and utilize the underutilized spectrum of the PN, an innovative approach with great potential to improve SE and optimize the use of available frequency resources [19].

Thus far, there exist scholarly researches that have explored the combination of CR with dual-hop links or NOMA. In CR-based RF-FSO dual-hop links, Neeraj et al. investigated amplify-and-forward (AF) in the presence of imperfect channel state information (CSI) [20]. Based on this, the case of multiple antennas at the relay (R), PT, and primary receiver (PR) was studied [21]. In addition, the multi-relay case introduced feedback links to determine and select the link with the best channel gain [22]. Notably, the fixed and CSI-assisted AF cases were examined to study single and two power-constrained points [23]. Shifting the focus to the fusion of CR and NOMA, CR-enabled AF-NOMA links in a two-user scenario were reviewed, considering the scaling coefficient of additive white gaussian noise (AWGN) [24]. Additionally, a CR-NOMA bidirectional cooperative relay scheme was proposed [25]. Lastly, a CR-based DF-NOMA in five modes was analyzed, considering the maximum transmit power of the ST and the maximum tolerable interference power of the

PN [26]. Through the above survey, it is found that most of the investigations only focus on a single problem, as CR with FSO/THz solves the problem of scarce spectrum resources and CR-NOMA improves SE; thus, it can be seen that the combination of both of them is essential and innovative, which arouses a great deal of research interest.

In this article, research gaps were identified, so FSO/THz-RF dual-hop links, and NOMA and CR were combined for their unique and fascinating characteristics. Consequently, a novel soft-switching FSO/THz-RF dual-hop AF-NOMA link based on CR is proposed. The link entails a soft-switching FSO/THz link serving as the first hop from the ST to the R, while the second hop exploits the potential of CR to effectively allocate the spectrum resources of the PN from the R to the secondary user (SU). The signals are transmitted via the AF-NOMA scheme, traversing an RF link to reach two users. In short, a soft-switching FSO/THz-RF dual-hop AF-NOMA link based on CR can use CR to address lower spectrum usage, NOMA can improve SE, the soft-switching FSO/THz pairing can reduce the impact of different atmospheric conditions, and RF can traverse obstacles to communicate with users.

The rest of the paper is organized as follows. Section 2 derives the probability density function (PDF) and cumulative distribution function (CDF) for FSO following the Weibull distribution, and the PDF and CDF for THz following the $\alpha - \mu$ distribution and considering pointing errors, and sets two FSO thresholds and a single THz threshold to form a soft-switching FSO/THz link. In Section 3, we first analyze the characteristics of the underlay CR and derive the relay transmission power with the maximum tolerable interference power in PN and the maximum relay transmission power. Subsequently, we analyze the properties of the NOMA system and derive the link OP by combining CR and NOMA. We deliver meaningful scenarios into the system to obtain beneficial numerical results in Section 4, and Section 5 concludes the paper.

## 2. System and Channel Models

The CR model comprises an ST, an R, two SUs, and a PU. Within the SN, the ST employs soft-switching FSO/THz links to propagate signals to R. As shown in Figure 1, R takes control of the spectrum band belonging to the PN using the CR and transmits signals to the two SUs via the AF-NOMA RF link. Without loss of generality, the channel gains are assumed to be $\left|h_{RU_1}\right|^2 < \left|h_{RU_2}\right|^2$ according to the inverse of the link distance because the greater the distance traveled, the greater the attenuation by atmospheric turbulence.

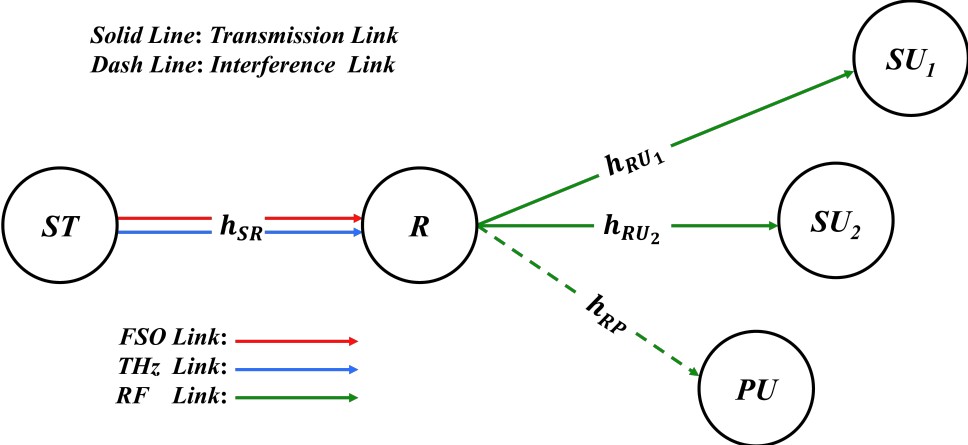

**Figure 1.** Soft-switching FSO/THz-RF dual-hop AF-NOMA link with CR.

### 2.1. Soft-Switching FSO/THz Link

The complementary characteristics of FSO and THz in coping with atmospheric conditions lead us to build hybrid FSO/THz links. Compared with hard-switching, soft-switching between FSO and THz links is used to avoid frequent switching and improve the lifetime of the links, and a more practical approach is to use two FSO thresholds and

a single THz threshold for soft-switching between the links. Therefore, in this section, we utilize the complementary characteristics of FSO and THz in different atmospheric environments to build a soft-switching FSO/THz scheme in the SR link. The exponential Weibull model is a commonly used model to characterize the atmospheric turbulence of FSO links in recent years, which not only can accurately simulate the optical irradiance under weak turbulence and strong turbulence, but also more importantly, the derivation of the exponential Weibull model is relatively simple, which makes it very suitable for the soft-switching case. Therefore, the exponential Weibull model is chosen in this paper, and the channel gain $x_F$ of the PDF is [27]:

$$f_{|h_F|^2}(x_F) = \frac{\alpha_F \beta_F}{\eta} \left(\frac{x_F}{\eta^2}\right)^{\frac{\beta_F-1}{2}} \exp\left[-\left(\frac{x_F}{\eta^2}\right)^{\frac{\beta_F}{2}}\right] \left\{1 - \exp\left[-\left(\frac{x_F}{\eta^2}\right)^{\frac{\beta_F}{2}}\right]\right\}^{\alpha_F-1}. \tag{1}$$

Subsequently, integrating (1) yields the CDF of the exponential Weibull model as

$$F_{|h_F|^2}(x_F) = \left\{1 - \exp\left[-\left(\frac{x_F}{\eta^2}\right)^{\frac{\beta_F}{2}}\right]\right\}^{\alpha_F}, \tag{2}$$

where $\alpha_F$ and $\beta_F$ are the shape parameter, and $\eta$ is the scale parameter.

For THz, channel fading is appropriately modeled as the $\alpha - \mu$ distribution, which is a generalized model that can be simplified to numerous important distributions such as Gamma, Nakagami-*m*, etc. [28]. THz link channel gain $x_T$ is affected by pointing errors; it follows $\alpha - \mu$ distribution as its PDF [29]:

$$f_{|h_T|^2}(\lambda_T) = \frac{g_T^2 \mu_T^{\frac{g_T^2}{\alpha_T}} \lambda_T^{\frac{g_T^2}{2}-1}}{2\left(A_{0,T} h_{l,T} \hat{h}_{f,T}\right)^{g_T^2} \Gamma(\mu_T)} \times$$

$$G_{1,2}^{2,0}\left[\frac{\mu_T}{\left(A_{0,T} h_{l,T} \hat{h}_{T,f}\right)^{\alpha_T}} \lambda_T^{\frac{\alpha_T}{2}} \middle| \begin{matrix} -;1 \\ 0, \frac{\alpha_T \mu_T - g_T^2}{\alpha_T}; - \end{matrix}\right]. \tag{3}$$

The CDF of $\alpha - \mu$ distribution obtained by integrating (3) and performing some algebraic calculations is

$$F_{|h_T|^2}(x_T) = \phi_T G_{2,3}^{2,1}\left[\psi_T x_T^{\frac{\alpha_T}{2}} \middle| \begin{matrix} 1;1+\frac{g_T^2}{\alpha_T} \\ \frac{g_T^2}{\alpha_T}, \mu_T; 0 \end{matrix}\right], \tag{4}$$

where $\phi_T = g_T^2 / [\alpha_T \Gamma(\mu_T)]$, $\psi_T = \mu_T / \left(A_{0,T} h_{l,T} \hat{h}_{f,T}\right)^{\alpha_T}$, and $G_{p,q}^{m,n}[\cdot]$ is a MeijerG function [30]. $g_T$ is the ratio between the equivalent beam radius and the pointing error displacement standard deviation at receive (RX). $A_{0,T}$ is the fraction of collected power at the origin. $h_{l,T}$ is the path loss of the THz link. $\alpha_T$ and $\mu_T$ denote the fading parameters, while $\hat{h}_{f,T}$ represents the $\alpha$-root mean value of the fading channel envelope.

Deriving the CDF of soft-switching FSO/THz-RF based on the CDF above for each link, the upper threshold $x_{th,U}^F$ and the lower threshold $x_{th,L}^F$ are set for the FSO link, and there is only one threshold, $x_{th}^T$, for the THz link, which is depicted in Figure 2 [31].

The specific operation of the soft-switching is based on the following process:

- If $x_F > x_{th,U}^F$, regardless of $x_T$, the FSO link is activated.
- If $x_{th,L}^F > x_F > x_{th,U}^F$, and given $x_F > x_{th,U}^F$ previously, regardless of $x_T$, the FSO link is activated.

- If $x_{th,L}^F > x_F > x_{th,U}^F$, and given $x_F < x_{th,L}^F$ previously, if $x_T > x_{th}^T$, the THz link is activated; if $x_T < x_{th}^T$, the hybrid link is outage.
- If $x_F < x_{th,U}^F$, if $x_T > x_{th}^T$, the THz link is activated; if $x_T < x_{th}^T$, the hybrid link is outage.

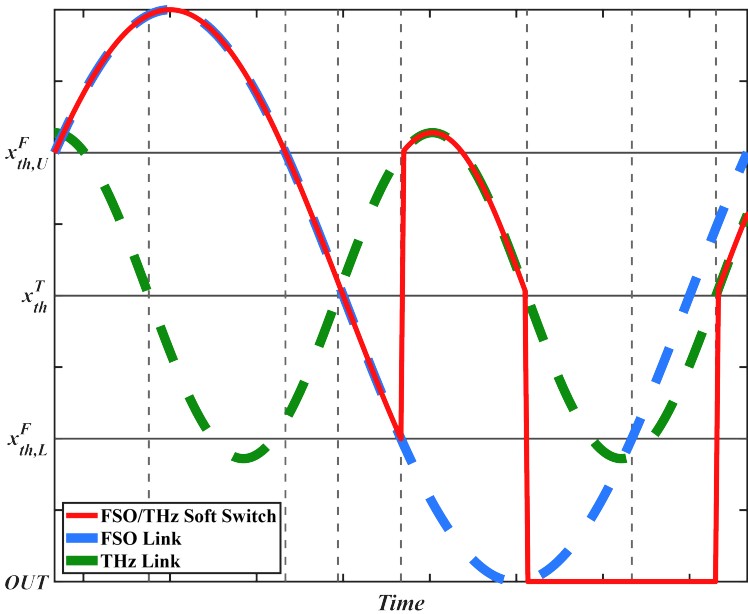

**Figure 2.** Description of the soft-switching FSO/THz hybrid link.

For ease of calculation, we introduce an offset $\Delta$, which is used to describe the offset of the two thresholds for the FSO link and the single threshold for the THz link from the center threshold such that $x_{th,U}^F = (1 + \Delta)x_{th}$, $x_{th,L}^F = (1 - \Delta)x_{th}$, and $x_{th}^T = x_{th}$. According to the specific operation of soft-switching, the OP of the FSO link in the soft-switching system can be expressed as

$$F_{|h_F|^2}^{Soft}(x_{th}) = P_{low}^F + P_{med}^F \left( \frac{P_{low}^F}{P_{low}^F + P_{hig}^F} \right), \tag{5}$$

where, $P_{low}^F = F_{|h_F|^2}((1 - \Delta)x_{th})$, $P_{med}^F = F_{|h_F|^2}((1 + \Delta)x_{th}) - F_{|h_F|^2}((1 - \Delta)x_{th})$, and $P_{hig}^F = 1 - F_{|h_F|^2}((1 + \Delta)x_{th})$.

Because THz has only a single threshold, the OP of the THz link in the soft-switching system can be expressed as

$$F_{|h_T|^2}^{Soft}(x_{th}) = F_{|h_T|^2}(x_{th}). \tag{6}$$

Therefore, in the case of soft-switching, the OP of the hybrid FSO/THz link can be expressed as

$$F_{|h_{SR}|^2}(x_{th}) = F_{|h_F|^2}^{Soft}(x_{th}) \times F_{|h_T|^2}^{Soft}(x_{th}). \tag{7}$$

*2.2. RF Link*

On the other hand, the PDF and CDF of the unordered variable of the channel gain $|h_{RU_n}|^2$ in the second slot follow the Nakagami-*m* distribution and can be represented as [32]

$$f_{|\tilde{h}_{RU_n}|^2}(x) = \frac{m_n^{m_n} x^{m_n - 1}}{\Omega_n^{m_n} \Gamma(m_n)} \exp\left( -\frac{m_n x}{\Omega_n} \right), \tag{8}$$

$$F_{|\widetilde{h}_{RU_n}|^2}(x) = \frac{Y\left(m_n, \frac{m_n x}{\Omega_n}\right)}{\Gamma(m_n)}, \tag{9}$$

where $m_n$ represents the fading coefficient and $\Omega_n = d_n^{-\alpha}$ is the average power with $d_n$ being the distance of the $n$th link and $\alpha$ the path loss exponent, $\Gamma(a) = \int_0^\infty x^{a-1}e^{-x}dx$ denotes the Gamma function, and $Y(m, x)$ is the lower incomplete gamma function. Specifically, $Y(a, x) = \int_0^x x^{a-1}e^{-x}dx$. Additionally, it is worth noting that $Y(a, x) = (a-1)!\left[1 - e^{-x}\sum_{i=0}^{a-1}(x^i/i!)\right]$ when $a$ is an integer greater than or equal to one. In this paper, we consider integer values for $m_n$; thus, we can rephrase the expression $F_{|\widetilde{h}_{RU_n}|^2}(x)$ as follows:

$$F_{|\widetilde{h}_{RU_n}|^2}(x) = 1 - \exp\left(-\frac{m_n x}{\Omega_n}\right)\sum_{r=0}^{m_n-1}\frac{\left(\frac{m_n x}{\Omega_n}\right)^r}{r!}. \tag{10}$$

With some algebraic calculations, the respective PDF and CDF for the ordered variable $|h_{RU_n}|^2$ can be written as [32]

$$f_{|h_{RU_n}|^2}(x) = \sum_{k=0}^{M-n}(-1)^k\binom{M-n}{k}f_{|\widetilde{h}_{RU_n}|^2}(x)\left[F_{|\widetilde{h}_{RU_n}|^2}(x)\right]^{n+k-1}, \tag{11}$$

$$F_{|h_{RU_n}|^2}(x) = Q\sum_{k=0}^{M-n}\frac{(-1)^k}{n+k}\binom{M-n}{k}\left[F_{|\widetilde{h}_{RU_n}|^2}(x)\right]^{n+k}, \tag{12}$$

where $Q = M!/[(M-n)!(n-1)!]$. By substituting the specific parameters of the two users, we can derive the corresponding PDF and CDF.

It is worth noting that the PN spectrum band is occupied in the second slot. Assuming the channel gain $|h_{RP}|^2$ follows a Rayleigh distribution, its CDF is

$$F_{|h_{RP}|^2}(x) = 1 - \exp\left(-\frac{x}{\Omega_P}\right), \tag{13}$$

where $\Omega_P = d_p^{-\alpha}$ holds the same significance as $\Omega_n$.

## 3. Performance Analysis

### 3.1. CR System

For the analysis in this section, the OP will be derived. The underlay CR we consider is the PowIntSCSI model with a constrained power budget at ST and computed statistical CSI at ST, which increases the practical application effect of the link [26]. Specifically, this model is restricted by both the maximum relay transmit power (i.e., $P_{Peak}$) and the maximum tolerable interference power of PN (i.e., $I$). In such a scenario, the quality of service (QoS) for the PU can be safeguarded through statistical constraints, ensuring that the probability of R causing interference to the PU surpassing the interference threshold $I$ remains below the predetermined threshold $\delta$. The transmitted power at R is defined as $P_R$, which is obtained as

$$\Pr\left(|h_{RP}|^2 P_R > I\right) \le \delta \implies 1 - F_{|h_{RP}|^2}\left(\frac{I}{P_R}\right) \le \delta \implies P_R \le \frac{I}{F_{|h_{RP}|^2}^{-1}(1-\delta)}, \tag{14}$$

where $F_{|h_{RP}|^2}^{-1}(x) = -\Omega_P\ln(1-x)$ represents the inverse function of $F_{|h_{RP}|^2}(x)$. Thus, the transmission power at R can be expressed as [26]

$$P_R = \min\left\{P_{Peak}, \frac{-I}{\Omega_P\ln\delta}\right\} = \begin{cases} P_{Peak}, & \text{if } P_{Peak} \le \frac{-I}{\Omega_P\ln\delta} \\ \frac{-I}{\Omega_P\ln\delta}, & \text{otherwise.} \end{cases} \tag{15}$$

In the subsequent derivation, $P_R$ will take the smaller value in (15) to represent the transmit power at $R$. As $I$ keeps increasing, $\frac{-I}{\Omega_P \ln \delta}$ will increase and $P_R$ will be equal to maximum relay transmission power $P_{Peak}$ at $R$.

### 3.2. NOMA System

In the NOMA system, it is challenging to consider a large number of users due to the occupancy of the PN spectrum band. Hence, we only focus on the superposition of signals from two users. Let us assume that the normalized signal $x_S = \sqrt{a_1 P_S} x_1 + \sqrt{a_2 P_S} x_2$. Here, $x_n$ is the signal required by the nth user, $P_S$ denotes the transmission power at ST, and $a_n$ is the power allocation coefficient (PAC) for the nth user, indicating the proportion of the transmitted power allocated to $x_n$ [32]. Following the principles of NOMA, $a_1 > a_2$ and $a_1 + a_2 = 1$. Therefore, the signal at $R$ is

$$y_R = h_{SR} \left( \sqrt{a_1 P_S} x_1 + \sqrt{a_2 P_S} x_2 \right) + n_R, \tag{16}$$

where $n_R \sim \mathcal{CN}(0,1)$ represents AWGN at $R$.

In the second slot, $R$ amplifies $y_R$ using the amplification coefficient $G$ before propagating it to all users. The signal received by the nth user can be expressed as

$$y_{U_n} = \sqrt{P_R} G h_{RU_n} y_R + n_{U_n}, \tag{17}$$

where $P_R$ represents the transmission power at $R$, and $n_{U_n} \sim \mathcal{CN}(0,1)$ denotes AWGN at the nth user. Thus, $G$ should be determined as

$$G = \frac{1}{\sqrt{\left( P_S |h_{SR}|^2 + 1 \right)}}. \tag{18}$$

For each user, the desired received signals are subject to interference from the signals of other users. Consequently, successive interference cancellation (SIC) should be performed at each user to obtain its own signal. The SIC decoding order is incrementally related to the user's channel gain $\left( |h_{RU_1}|^2 < |h_{RU_2}|^2 \right)$. Therefore, at the nth user, it will detect the jth user's signal where $j < n$, and then subtract it from the nth user's received signal sequentially. The signal of the mth user with $m > n$ will be treated as noise at the nth user [32].

The generalized formula for the signal-to-interference plus noise ratio (SINR) can be computed as

$$\gamma_{j \to n} = \frac{a_j P_S P_R |h_{SR}|^2 |h_{RU_n}|^2}{P_S P_R |h_{SR}|^2 |h_{RU_n}|^2 \sum\limits_{i=j+1}^{M} a_i + P_S |h_{SR}|^2 + P_R |h_{RU_n}|^2 + 1}, \tag{19}$$

where $\gamma_{j \to n}$ represents the SINR of the nth user to decode the signal from the jth user.

### 3.3. Outage Probability

Hence, a user achieves successful signal reception by decoding and eliminating the signal from users with higher PAC and successfully decoding its own signal. Based on this, the OP can be derived as

$$P_{OUT}^n = 1 - \Pr\left(\gamma_{j \to n} > \gamma_{th,j}\right)$$

$$= 1 - \Pr\left(\frac{a_j P_S P_R |h_{SR}|^2 |h_{RU_n}|^2}{P_S P_R |h_{SR}|^2 |h_{RU_n}|^2 \sum\limits_{i=j+1}^{M} a_i + P_S |h_{SR}|^2 + P_R |h_{RU_n}|^2 + 1} > \gamma_{th,j}\right)$$

$$= 1 - \Pr\left(\left[P_S P_R \left(a_j - \gamma_{th,j} \sum\limits_{i=j+1}^{M} a_i\right) |h_{RU_n}|^2 - P_S \gamma_{th,j}\right] |h_{SR}|^2 > \gamma_{th,j}\left(P_R |h_{RU_n}|^2 + 1\right)\right)$$

$$= 1 - \Pr\left(|h_{RU_n}|^2 > \frac{\gamma_{th,j}}{P_R\left(a_j - \gamma_{th,j}\sum\limits_{i=j+1}^{M} a_i\right)} \triangleq \theta_j, |h_{SR}|^2 > \frac{\theta_j\left(1 + P_R |h_{RU_n}|^2\right)}{P_S\left(|h_{RU_n}|^2 - \theta_j\right)}\right),$$

(20)

where $\gamma_{th,j}$ denotes the threshold of the $j$th user and $\theta_j$ is some constant product of the $j$th user.

After some algebraic manipulations, the expression can be written as

$$P_{OUT}^n = \int_0^{\theta_n^*} f_{|h_{RU_n}|^2}(x)dx + \int_{\theta_n^*}^{\infty} f_{|h_{RU_n}|^2}(x) F_{|h_{SR}|^2}\left(\frac{\theta_n^*\left(1 + P_R |h_{RU_n}|^2\right)}{P_S\left(|h_{RU_n}|^2 - \theta_n^*\right)}\right)dx,$$

(21)

where $\theta_n^* = \max(\theta_j, \cdots, \theta_n)$. Substituting (7), (12), and (15) into (21) yields the OP for the corresponding user.

## 4. Numerical Results

In this section, we provide meaningful values to perform a numerical analysis of the proposed link. Unless otherwise specified, the NOMA system considered here has $M = 2$, $a_1 = 0.51$, $a_2 = 0.49$, $\gamma_{th,1} = 51/149$, $\gamma_{th,2} = 49/100$, and $\Delta = 5\%$. Regarding the frequency and wavelength, the RF frequency is taken as 60 GHz, the THz frequency is 0.1 THz, and the FSO wavelength is 1550 nm [31]. For atmospheric turbulence and pointing error scenarios, we set the moderate turbulence parameters as $\alpha_F = 1$, $\beta_F = 2$, $\eta = 2/\sqrt{\pi}$, $\alpha_T = 1.7$, $\mu_T = 1.5$, $m_1 = 1$, and $m_2 = 1$. For weak turbulence, the parameters are $\alpha_F = 1$, $\beta_F = 3$, $\eta = 2/\Gamma(4/3)$, $\alpha_T = 1.7$, $\mu_T = 1.7$, $m_1 = 2$, and $m_2 = 2$. The large pointing error is represented by $g_T = 1.6224$, whereas the small pointing error is denoted by $g_T = 2.9203$. The path loss exponent $\alpha = 2$, with distances $d_{SR} = 200$ m, $d_{RU_1} = 20$ m, $d_{RU_2} = 20$ m, and $d_{RP} = 5$ m. In the subsequent analysis, we primarily focus on the performance of OP under various conditions.

Figure 3 investigates the relationship between OP and PAC under different $I$ in weak turbulence. We set $\gamma_{th,j} = (a_j \rho_m)/\left(1 + \rho_m \sum_{i=j+1}^{M} a_i\right)$, where $\rho_m = \text{floor}[\rho + 1]$ and $\rho$ is a value on the ordinate of the function $\rho\left(\gamma_{th,j}\right) = \gamma_{th,j}/\left(a_j - \gamma_{th,j}\sum_{i=j+1}^{M} a_i\right)$, and floor$[\cdot]$ is rounded down. The results demonstrate that the link performance improves as the mean deviation of PAC decreases. Additionally, due to variations in channel gains, User2 outperforms User1. Comparing different $I$, we observe that a larger $I$ leads to better link performance, benefiting from the ability of the PN to withstand greater interference. Therefore, we consistently adopt the optimal PAC and SINR thresholds throughout the analysis.

In Figure 4, numerous deductions may be discerned. Firstly, with the elevation of $I$, the link's performance exhibits enhancement. Nevertheless, owing to the constraints imposed by both $I$ and $P_{Peak}$ upon $P_R$, the link's performance reaches a state of equilibrium at a specific $I$, as determined by (13). Furthermore, it is noteworthy that augmenting both $P_S$ and $P_{Peak}$ can amplify the link's performance, albeit the enhancement brought about by increasing $P_{Peak}$ is notably more pronounced. This discrepancy arises from the operation

of the link using AF-NOMA, wherein $P_S$ contends with more extended link distances compared to $P_{Peak}$. Consequently, for the ensuing analysis, we establish $P_S = 15$ dBm and $P_{Peak} = 40$ dBm.

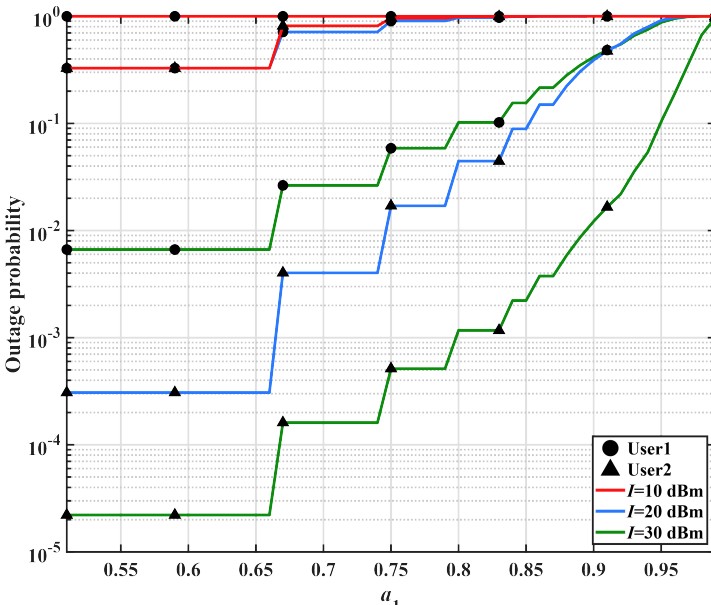

**Figure 3.** The relationship between OP and PAC under different $I$ in weak turbulence.

The comparison between (a), (c), and (e) with (b), (d), and (f) in Figure 5 reveals that User2 outperforms User1 due to the difference in channel gains. Moreover, by comparing the three columns (a) (b), (c) (d), and (e) (f), it is evident that the link performance increases with the increase in $P_S$. Finally, we observe that a single change in either $P_{Peak}$ or $I$ does not significantly enhance the link performance. Only when both parameters are increased simultaneously is the link performance improved. Thus, the performance advantage is concentrated in the rectangular region in the upper right corner of the figure.

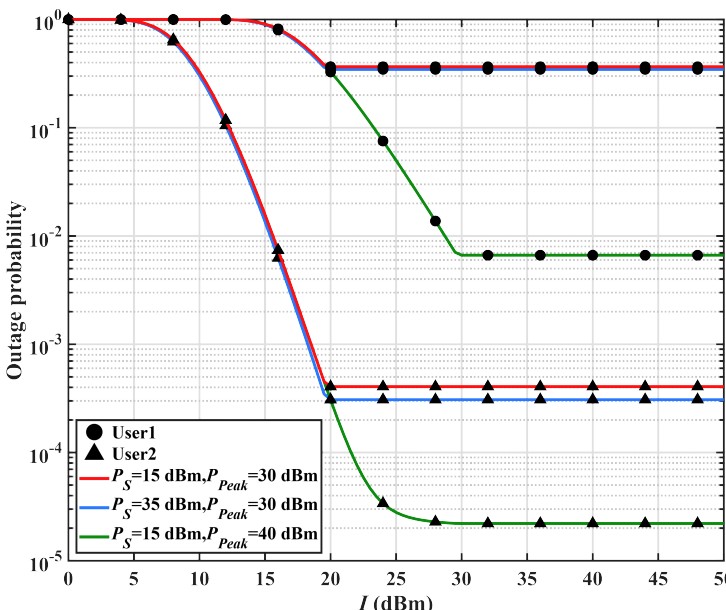

**Figure 4.** The impact of different values of $P_S$ and $P_{Peak}$ on OP under weak turbulence.

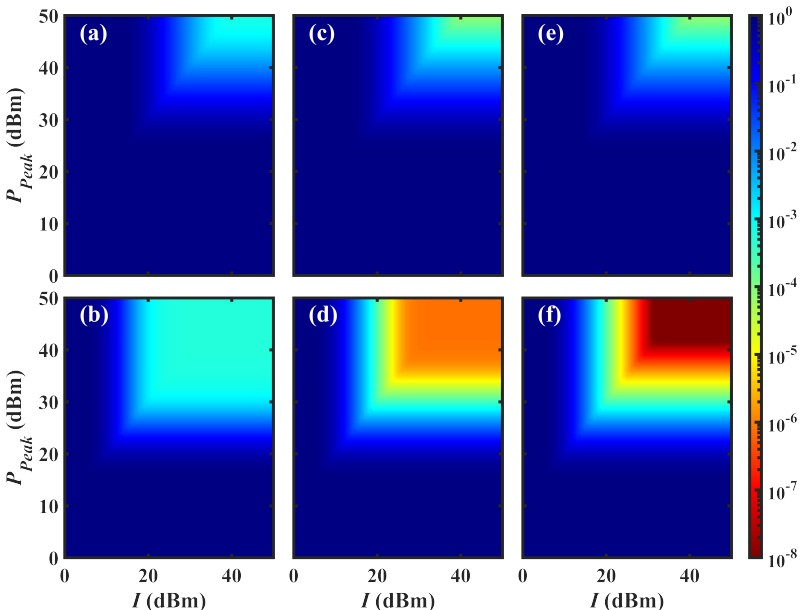

**Figure 5.** A comparison of $P_{Peak}$ and $I$ on OP under different $P_S$. (**a**,**b**) correspond to $P_S = 10$ dBm, (**c**,**d**) correspond to $P_S = 20$ dBm, and (**e**,**f**) correspond to $P_S = 30$ dBm. In (**a**,**c**,**e**), User1 is represented, while in (**b**,**d**,**f**), User2 is represented.

Figure 6 depicts three atmospheric scenarios: Case1 involves moderate turbulence with large pointing errors, Case2 features weak turbulence with large pointing errors, and Case3 exhibits weak turbulence with small pointing errors. Upon comparing these scenarios, it becomes evident that the link performance is superior in the cases of weak turbulence and small pointing errors. Specifically, around the smoothing point at $I = 35$ dBm, User2 achieves an OP of $4.4482 \times 10^{-4}$ in Case1, $4.7948 \times 10^{-5}$ in Case2, and $9.3066 \times 10^{-6}$ in Case3.

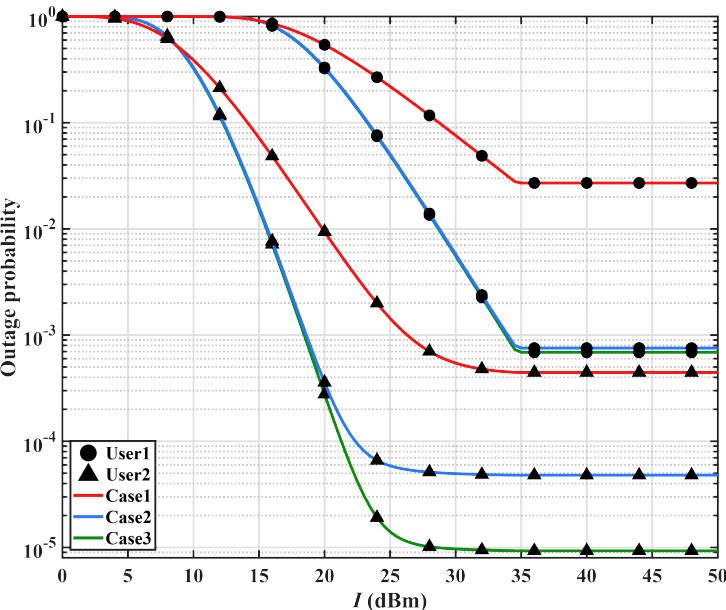

**Figure 6.** Impact of different turbulence and pointing errors on link performance.

In Figure 7, we maintain a constant SN power, denoted as $P_S = P_{Peak}$. We compare the OP at different $I$ and contrast them with the scenario without CR. As the $I$ increases, it signifies a higher tolerance to interference, leading to improved link performance. Thus, we observe a consistent performance in the low-power region in all cases. The key distinction

lies in the performance that is constrained by both $P_{Peak}$ and $I$ with CR, whereas without CR, it is solely limited by $P_{Peak}$. The results reveal that when $I = 35$ dBm and the SN power ranges from 0 to 50 dBm, there is minimal disparity in performance between the with-CR and without-CR scenarios. Encouragingly, even as $I$ continues to rise, the performance discrepancy remains insignificant.

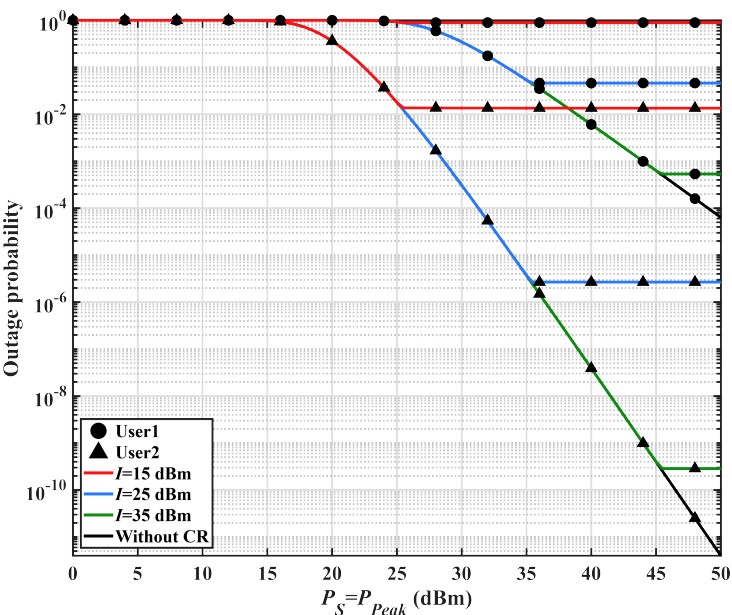

**Figure 7.** A comparison of the power of the SN network and OP at different $I$.

## 5. Conclusions

This paper introduces an innovative AF-NOMA-assisted soft-switching FSO/THz-RF double-hop link based on CR, addressing the scarcity of spectrum resources through FSO and THz. Firstly, a soft-switching FSO/THz system was built on the SR link, and both of them were paired with each other to largely mitigate the impact of atmospheric conditions on the SR link. In addition, NOMA allows signals of different powers to be superimposed on an identical frequency band, and CR allows that frequency band to be present in the main network, thus resolving frequency band resources and improving SE. The study primarily focuses on three key factors impacting link performance: $P_S$, $P_{Peak}$, and $I$. Notably, $P_S$ has a relatively minor influence on performance improvement due to its longer propagation distance. Additionally, the SN power is maintained at a constant level compared with the case without CR. It is evident that the performance without CR surpasses that with CR, mainly due to the imposed constraints. Nevertheless, the incorporation of CR significantly enhances SE and spectrum usage. The numerical results demonstrate the superiority of this link, which offers a fresh solution for next-generation communications.

**Author Contributions:** Conceptualization, methodology, R.L. and H.L.; software, validation, writing—original draft preparation, R.L.; formal analysis, investigation, data curation, R.W. and Z.W. (Zhongchao Wei); writing—review and editing, visualization, Z.W. (Ziyang Wang), X.W., J.L., Y.W., and Y.Z.; supervision, project administration, funding acquisition, H.L. All authors have read and agreed to the published version of the manuscript.

**Funding:** This research was funded by the National Natural Science Foundation of China (No. 62175070 and 61875057), the Guangdong Basic and Applied Basic Research Foundation (No. 2021A151 5012652 and 2022A1515110752), and the Science and Technology Program of Guangzhou (No. 202201010340, and 2019050001).

**Institutional Review Board Statement:** Not applicable.

**Informed Consent Statement:** Not applicable.

**Data Availability Statement:** Not applicable.

**Conflicts of Interest:** The authors declare no conflict of interest.

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
