# Peer review of "Performance Analysis of Soft-Switching FSO/THz-RF Dual-Hop AF-NOMA Link Based on Cognitive Radio"

_photonics, doi:10.3390/photonics10101086_

Round 1
Reviewer 1 Report

The manuscript is well-presented, but there are certain areas that could be improved to make the language more natural and idiomatic.
Reviewer 2 Report
The paper describes and analyzes a free-space optical (FSO) link with cognitive radio (CR) capability that applies a dual-hop amplify-and-forward (AF) non-orthogonal multiple access (NOMA) technique. The aim of the proposed link is to address the scarcity of spectrum resources and, at the same time, achieve higher data ranges by utilizing the THz/optical part of the spectrum.
The novelty of the article mainly lies in the combination of the THz/optical part of the spectrum with the CR concept and the AF/NOMA techniques.
The article provides a concrete mathematical analysis and a fair amount of simulation results that support its case and conclusions. The list of references is adequate and most of the references are recent.
The article is publishable subject to comments shown below:
• I do not understand the statement in lines 22-24. The authors refer to the 30MHz – 3GHz spectrum (which includes much lower frequencies that the THz/optical one) and claim that the usage of this part (30MHz – 3GHz) is particularly low. If this is the case, why associate the use of the THz/optical spectrum with the scarcity of the spectrum? Then a proper argument would be that the primary reason for utilizing the THz/optical range is increasing the bandwidth and the bit-rate of the links and, at the same time, provide a much wider free-of-license frequency range.
• The authors should be more specific regarding the frequency range they propose to be used (THz RF / optical are rather vague terms). Besides, I am not sure that THz RF is an acceptable term since a frequency of 1THz (wavelength equal to 100 μm) is in the infrared (IR) range.
• The article is generally well written, however some editing regarding the use of English (mainly style) is necessary.
The article is generally well written, however some editing regarding the use of English (mainly style) is necessary.
Reviewer 3 Report
Performance Analysis of Soft-Switch FSO/THz-RF Dual-hop AF-NOMA Link Based on Cognitive Radio
I am deeply interested by the analysis of the FSO/THz-RF dual-hop link with CR-NOMA. I believe it can serve as a valuable reference for future communication systems. I have a few suggestions to offer below.
In this paper, the performance of AF Soft-switch FSO/THz-RFdual-hop CR-NOMA communication systems has been analytically investigated. The research topic under investigation is new, the results presented are interesting, while some useful insights have been also reported. I believe it can serve as a valuable reference for future communication systems. Howerver, it seems that some viewpoints based on CR taht is not well clear. Next a list of comments that should be taken into account is presented.
1.I believe you can augment the introduction and pertinent conclusions to meet the article's word count requirements.
2.What advantages do you perceive in your utilization of underlay CR compared to interweave CR and overlay CR?
3.From equations (13) and (18), it can be observed that there is a connection between the three crucial parameters, maximum tolerable interference power in primary network , transmit power in secondary transmitter and maximun relay transmission power , in your paper. However, how do these parameters specifically influence the link's outage probability?
4.Upon scrutinizing your Figure 3, it becomes apparent that there exists a smooth interval under higher conditions. However, the inflection points for the three scenarios differ. I would like to inquire about the parameters governing the determination and the underlying principles behind these inflection points.
Minor editing of English language required
Round 2
Reviewer 2 Report
I am satsfied with the response of the reviewers' revisions regarding my report.
I consider the article publisable subject to minor editing regarding the use of English (that it will be done, anyway, during the publication process).
A minor editing regarding the use of English (that it will be done, anyway, during the publication process)is necessary.